# Enhanced Cardiorenal Protective Effects of Combining SGLT2 Inhibition, Endothelin Receptor Antagonism and RAS Blockade in Type 2 Diabetic Mice

**DOI:** 10.3390/ijms232112823

**Published:** 2022-10-24

**Authors:** Ander Vergara, Conxita Jacobs-Cacha, Carmen Llorens-Cebria, Alberto Ortiz, Irene Martinez-Diaz, Nerea Martos, Pamela Dominguez-Báez, Mireia Molina Van den Bosch, Sheila Bermejo, Michael Paul Pieper, Begoña Benito, Maria Jose Soler

**Affiliations:** 1Nephrology and Kidney Transplantation Research Group, Vall d’Hebron Institut de Recerca (VHIR), Passeig Vall d’Hebron 119-129, 08035 Barcelona, Spain; 2Nephrology Department, Vall d’Hebron Hospital Universitari, Passeig Vall d’Hebron 119-129, 08035 Barcelona, Spain; 3IIS-Fundación Jiménez Diaz, Fundación Renal Iñigo Álvarez de Toledo-IRSIN, REDinREN, Instituto de Investigación Carlos III, Universidad Autónoma de Madrid, Av. de los Reyes Católicos 2, 28040 Madrid, Spain; 4Cardio-Metabolic Diseases Research, Boehringer Ingelheim Pharma GmbH & Co. KG, Birkendorfer Str. 65, 88397 Biberach an der Riß, Germany; 5Cardiology Research Group, Vall d’Hebron Institut de Recerca (VHIR), Vall d’Hebron Hospital Universitari, Vall d’Hebron Barcelona Hospital Campus, Passeig Vall d’Hebron 119-129, 08035 Barcelona, Spain; 6Cardiology Department, Vall d’Hebron Hospital Universitari, Passeig Vall d’Hebron 119-129, 08035 Barcelona, Spain; 7Deparment of Medicine, Universitat Autònoma de Barcelona, Av. de Can Domènech, 08193 Bellaterra, Spain

**Keywords:** diabetes, diabetic nephropathy, chronic kidney disease, sodium–glucose cotransporter 2 inhibitors, endothelin receptor antagonists

## Abstract

Treatments with sodium–glucose 2 cotransporter inhibitors (SGLT2i) or endothelin receptor antagonists (ERA) have shown cardiorenal protective effects. The present study aimed to evaluate the cardiorenal beneficial effects of the combination of SGLT2i and ERA on top of renin–angiotensin system (RAS) blockade. Type 2 diabetic mice (db/db) were treated with different combinations of an SGLT2i (empagliflozin), an ERA (atrasentan), and an angiotensin-converting enzyme inhibitor (ramipril) for 8 weeks. Vehicle-treated diabetic mice and non-diabetic mice were included as controls. Weight, blood glucose, blood pressure, and kidney and heart function were monitored during the study. Kidneys and heart were collected for histological examination and to study the intrarenal RAS. Treatment with empagliflozin alone or combined significantly decreased blood glucose compared to vehicle-treated db/db. The dual and triple therapies achieved significantly greater reductions in diastolic blood pressure than ramipril alone. Compared to vehicle-treated db/db, empagliflozin combined with ramipril or in triple therapy significantly prevented GFR increase, but only the triple combination exerted greater protection against podocyte loss. In the heart, empagliflozin alone or combined reduced cardiac isovolumetric relaxation time (IVRT) and left atrium (LA) diameter as compared to vehicle-treated db/db. However, only the triple therapy was able to reduce cardiomyocyte area. Importantly, the add-on triple therapy further enhanced the intrarenal ACE2/Ang(1-7)/Mas protective arm of the RAS. These data suggest that triple therapy with empagliflozin, atrasentan and ramipril show synergistic cardiorenal protective effects in a type 2 diabetic mouse model.

## 1. Introduction

Diabetes is the leading acquired risk factor for an accelerated progression of CKD to kidney failure [1]. Around 20 per cent of incident patients on kidney replacement therapy reach this stage due to diabetic kidney disease (DKD) [2]. Since the EMPA-REG OUTCOME clinical trial in 2016 [3], sodium–glucose cotransporter 2 inhibitors (SGLT2i) have repeatedly been shown to delay DKD progression in patients with preserved [4,5] and decreased glomerular filtration rates (GFR) [4,6] when added on top of renin–angiotensin system (RAS) blockade. SGLT2i represents the first drug class that has delayed DKD progression since the beneficial effects of RAS blockers were demonstrated [7,8]. More recently, selective endothelin receptor subtype A antagonists (ERA) [9] reduced albuminuria and delayed DKD progression [10]. Atrasentan, which has a selectivity for type A versus B receptor of 1200:1, is one of the best studied ERAs [11]. The SONAR trial demonstrated that atrasentan improved the composite renal outcome of doubling of serum creatinine or kidney failure after a median follow-up of 2.2 years when compared to patients treated with placebo [11]. However, as previously observed for avosentan [12], fluid retention, and anemia were more frequent in the atrasentan arm [11]. This favors heart failure and limits the use of ERAs in clinical practice. In fact, ERAs are not available for routine treatment of DKD. In contrast, SGLT2i displays diuretic effects [13], increases hemoglobin levels [14], and decreases the risk of heart failure [15,16]. For this reason, the combination of an SGLT2i with ERA may improve the safety of the latter group [17]. In addition, considering the different mechanisms of action, combining an SGLT2i with ERA may provide synergistic cardiorenal protective effects. Recently, a post hoc analysis of the SONAR trial in patients with type 2 diabetes and chronic kidney disease observed that the combination of atrasentan and SGLT2i was beneficial in terms of less fluid retention and a larger decrease in albuminuria compared to ERA alone [17].

In the present study, we analyzed the renal and heart protective effects of combining ERA and SGLT2i on top of RAS blockade in the db/db mouse, which is a widely used rodent model for type 2 diabetes research. Our main outcome was to study the effect of these drugs on early diabetic hyperfiltration, and our secondary outcomes were to identify kidney and heart protective effects at the tissue level. Moreover, we studied the impact of ERAs and SGLT2i on the intrarenal RAS, a pathway clearly involved in the development of diabetic nephropathy [18].

## 2. Results

### 2.1. Empagliflozin, Atrasentan, Ramipril, or Their Combinations Did Not Modify Body Weight or Body Fat

After 8 weeks of treatment, vehicle-treated db/db mice weighed 13.0 g (95% CI: 9.6–16.3) more than vehicle-treated db/m mice (Table 1). No differences were found between vehicle-treated db/db mice and mice treated with empagliflozin, ramipril, atrasentan, or their combinations, except for the ATR + RAM group. The latter group showed a decrease in body weight not accompanied by a reduction in body fat volume (Table 1), suggesting that the effect is unrelated to the study treatment. In line with the higher body weight, db/db mice also showed a higher body fat volume (18.4 cm^3^, IQR: 16.1–23.5) than vehicle-treated db/m mice (3.3 cm^3^, IQR: 1.9–4.9) (Table 1). Thus, body fat was 39.5% (95% CI: 32.3–46.7) higher in db/db mice than in db/m mice when values were adjusted to body volume. Further, only the group treated with the triple therapy showed increased subcutaneous and intra-abdominal fat volume as compared to the vehicle-treated db/db despite the body weight not being significantly higher in this group (Table 1).

### 2.2. Empagliflozin Alone or in Combination Reduced Blood Glucose

After 8 weeks of treatment, vehicle-treated db/db developed hyperglycemia (394 mg/dL, 95% CI: 359–429) (Figure 1A,B). Only empagliflozin, alone or in combination, reduced fasting blood glucose (mean reduction of 213 mg/dL in the 3 groups) in db/db mice (Figure 1B). Thus, empagliflozin preserves its blood-glucose-lowering effect under dual or triple therapy combinations. In addition, empagliflozin, in monotherapy or combined, increased glucosuria (*p* < 0.001 for empagliflozin’s main effect) (Figure 1C). However, this effect was milder in the EMP + RAM + ATR group.

### 2.3. Dual or Triple Combination Therapy Further Reduced Blood Pressure

Although no differences were observed in BP before starting the treatments (Appendix A), at the end of the experiment, systolic and diastolic BP were slightly lower in vehicle-treated db/db than in non-diabetic controls (Figure 1D,E). Ramipril in monotherapy significantly reduced systolic BP by 13.2 mmHg (95% CI: 4.8–21.6) compared with vehicle-treated db/db mice, while empagliflozin or atrasentan alone had no impact on systolic or diastolic BP. Ramipril in combination with empagliflozin or atrasentan reduced both systolic and diastolic BP as compared with vehicle-treated db/db mice, which was not observed with ramipril alone (Figure 1D,E). Thus, combination therapy offers a larger decrease in BP than ramipril alone, but triple therapy does not further reduce BP as compared with dual therapy. No evident differences were observed in heart rate (Figure 1F).

### 2.4. Empagliflozin Combined with Ramipril or Triple Therapy Prevent Diabetic Glomerular Hyperfiltration

Albuminuria and GFR were measured as surrogate markers of diabetic nephropathy. Consistent with the development of hyperfiltration, GFR increased in db/db mice after 8 weeks on vehicle Figure 2A and Appendix A) compared to non-diabetic db/m mice. Monotherapy approaches with either ramipril, empagliflozin, or atrasentan did not prevent hyperfiltration (Figure 2A), although ramipril’s main effect was significant (*p* = 0.005), suggesting that the ACEi component has an impact in GFR modulation. Interestingly, the combination of empagliflozin and ramipril as well as the triple therapy prevented hyperfiltration (92 ± 311 µL/100 g/min increase from baseline in vehicle-treated db/db mice vs. 192 ± 335 µL/100 g/min decrease in db/db mice treated with the combination of empagliflozin and ramipril, *p* = 0.037, and 189 ± 407µL/100 g/min decrease in the mice treated with the triple therapy, *p* = 0.025) (Figure 2A). Additionally, water intake was maintained in all the groups at the end of the treatment (Appendix A). Thus, dehydration is an unlikely cause of the reduction in GFR observed in these two groups.

Albuminuria increased in vehicle-treated diabetic mice (1575 µg/mg, 95% CI: 922–3268, vs. db/m mice) (Figure 2B). However, there was a large variability in albuminuria levels and despite a trend towards a reduced albuminuria for all active treatment arms, it only reached statistical significance for empagliflozin monotherapy (UACR 897 µg/mg, 95% CI: 216–2131) (Figure 2B).

### 2.5. Combination Therapy with Empagliflozin, Ramipril, and Atrasentan Increased Protective Effects against Diabetic Kidney Injury

Diabetic nephropathy was evaluated in kidney histology by measuring glomerular mesangial matrix area using PAS staining and by evaluating podocyte density. Mesangial matrix expansion was observed in vehicle-treated db/db mice (7.9%, 95% CI: 5.5–10.0, higher mesangial matrix area than non-diabetic mice), consistent with histological diabetic nephropathy (Figure 3A,B). Empagliflozin, ramipril, and atrasentan, alone or in combination, significantly reduced mesangial matrix expansion in db/db mice (Figure 3B). Thus, dual or triple therapy did not further add to the effect of monotherapy in this regard. In addition, vehicle-treated db/db mice showed a reduced podocyte density when compared to non-diabetic controls. All treatments except atrasentan monotherapy were able to recover podocyte density. However, the triple therapy showed the greatest increase in podocyte density (754 podocytes per mm^2^ (IQR: 607–814) in the vehicle-treated db/db vs. 914 podocytes per mm^2^ (IQR: 855–1048) in db/db EMP + ATR + RAM) (Figure 3C,D). Additionally, empagliflozin slightly improved renal tubulointerstitial collagen deposition (*p* = 0.017 for empagliflozin’s main effect), although this trend was not identified in all the groups receiving empagliflozin (Appendix A).

### 2.6. Atrasentan Alone Did Not Improve Diastolic Dysfunction, but Did Not Interfere with Protection Exerted by Ramipril or Empagliflozin

Vehicle-treated diabetic mice showed incipient signs of diastolic dysfunction after 8 weeks (Table 2 and Appendix A). This was observed by higher LA diameter (2.29 mm, IQR: 2.15–2.49) and longer isovolumetric relaxation times (IVRT, 19.9 ms, IQR: 18.2–20.8) than non-diabetic littermates (1.83 mm, IQR: 1.75–2.06, and 14.3 ms, IQR: 12.5–17.0, respectively). Ramipril in monotherapy or combined with atrasentan improved IVRT in db/db mice (*p* = 0.039 for ramipril’s main effect) but did not modify the LA diameter (Table 2). Only empagliflozin monotherapy or combined with ramipril as dual or triple therapy decreased both IVRT and LA diameter in db/db mice. Thus, both ramipril and empagliflozin provide cardiac protection, although this was more comprehensive for empagliflozin. Triple therapy did not further add cardiac structure and function protection, but its atrasentan component did not detract from it either.

### 2.7. Only Triple Therapy Prevents Cardiomyocyte Hypertrophy While Different Interventions Protect against LV Fibrosis

Cardiomyocyte hypertrophy was evaluated in H&E-stained LV sections. An increased cardiomyocyte area was observed in vehicle-treated db/db mice (142 µm^2^, IQR: 117–156, in the db/db group vs. 106 µm^2^, IQR: 98–127, in non-diabetic controls), consistent with cardiomyocyte hypertrophy (Figure 4A,B). Only triple therapy significantly decreased the size of cardiomyocytes (124 µm^2^, IQR: 103–132, *p* = 0.031 vs. vehicle-treated db/db mice) (Figure 4B).

Histological results were consistent with those of LV myosin heavy-chain gene expression. Diabetes was associated with unchanged α-myosin heavy-chain (α-MHC) (Figure 4E) but increased β-myosin heavy-chain (β-MHC) gene expression (2.8-fold, IQR: 1.2–6.9, increase vs. db/m mice, *p* = 0.042) (Figure 4F). All interventions except atrasentan decreased β-MHC. Additionally, dual or triple therapy increased the α-MHC/β-MHC ratio, preventing the isoform shift to a more economical myosin heavy isoform (Figure 4G).

Further, myocardial fibrosis measured by Picrosirius red staining was increased in vehicle-treated diabetic mice (Figure 4C). All active treatments protected against collagen accumulation (Figure 4C,D). In this regard, there was an increase (1.5-fold, IQR: 0.9–1.7) in LV collagen I gene expression in vehicle-treated db/db mice with an almost significant diabetes’ main effect (*p* = 0.065) (Figure 4H). All interventions either in monotherapy or in combination lead to significantly lower LV collagen I gene expression, except for the combination of empagliflozin and ramipril (Figure 4H).

### 2.8. Ramipril Activated the Intrarenal ACE2/Ang(1-7)/Mas Pathway and This Is Maximized by Triple Therapy

To evaluate the impact of interventions on the intrarenal RAS, first, we analyzed the kidney gene expression of angiotensinogen (AGT), which was 2.3-fold (IQR: 1.6–3.6) higher in vehicle-treated db/db mice than in db/m (Figure 5A). This increase in renal AGT gene expression correlated with increased urine AGT excretion in the vehicle-treated db/db as compared to their non-diabetic littermates (Figure 5B). These findings depict an increased renal expression of the main precursor of RAS in a diabetic milieu. The triple therapy significantly reduced AGT mRNA expression but increased AGT urine excretion (Figure 5A,B). In this sense, only diabetes’ main effect was significant (*p* < 0.001 in both cases) suggesting that these variations are not related to the triple therapy effect. No relevant differences were observed in serum AGT levels (Figure 5C).

Renin mRNA expression was 2.1-fold higher (IQR: 1.1–3.9) in vehicle-treated db/db mice. Renin expression was further increased in db/db mice treated with ramipril alone (7.6-fold, IQR: 4.8–14.4, higher than in db/m mice) and was even higher in mice treated with ramipril as part of dual or triple therapy combinations (3.5-fold, IQR: 2.3–4.6, 3.5-fold, IQR: 2.4–5.1 and 2.3-fold, IQR: 1.0–4.1, higher in the EMP + RAM, EMP + RAM + ATR and ATR + RAM groups, respectively, when compared to db/db mice treated with ramipril alone) (Figure 5D). Assessment of renin protein by immunohistochemistry was concordant with gene expression results (Figure 5E,F). The juxtaglomerular renin immunostaining area was 140 µm^2^ (95% CI: 58–223) higher in db/db mice treated with ramipril than in vehicle-treated db/db mice. Moreover, dual or triple therapies, including ramipril further increased renin immunostaining, reaching a statistically significant difference for the combination of ramipril and atrasentan (167 µm^2^, 95% CI: 84–249 higher than ramipril alone) and for the triple therapy (298 µm^2^, 95% CI: 163–433 higher than ramipril alone) (Figure 5E,F).

ACE and ACE2 gene expression and activity were evaluated in cortical kidney extracts, and the ACE2/ACE gene expression and enzymatic activity ratio was calculated to assess whether the classical or the non-classical RAS pathway was the predominant RAS axis. Non-diabetic vehicle-treated db/m mice were the reference group. In vehicle-treated db/db mice, the ACE2/ACE mRNA ratio was 5.6-fold (IQR: 3.8–9.5) higher than in db/m mice and remained high in treated mice (Figure 6A). In agreement with the gene expression data, the ACE2/ACE activity ratio was also higher in vehicle-treated diabetic mice (3.1-fold, IQR: 1.5–10.2) than in db/m mice (Figure 6B). The changes in the ratio observed in vehicle-treated db/db were mainly due to a decrease in ACE expression rather than an increase in ACE2 (Appendix A). As expected, ramipril in monotherapy or in combination reduced ACE activity when compared to vehicle-treated db/db mice (Appendix A). Therefore, the ACE2/ACE activity ratio was increased in db/db mice receiving ramipril alone compared to vehicle-treated db/db mice (7.3-fold (IQR: 5.6–18.1) vs. 3.1-fold (IQR: 1.5–10.2) in db/db RAM and db/db groups, respectively). In addition, the combination of ramipril with empagliflozin and/or atrasentan further increased the ACE2/ACE activity ratio over ramipril alone, showing in the three groups a statistically significant difference vs. the db/db group (Figure 6B). Interestingly, a similar ACE2/ACE activity ratio was observed in serum with the highest increase of ACE2 over ACE obtained by the combined therapies of EMP + RAM, EMP + RAM + ATR and ATR + RAM (Figure 6C).

Thus, diabetic nephropathy was characterized by increased kidney angiotensinogen expression, evidencing activation of intrarenal RAS. Ramipril-containing interventions increased renin and the ACE2/ACE ratio, but the triple therapy resulted in the highest renin and ACE2/ACE ratio, evidencing activation of the non-classical RAS pathway and the ACE2/Ang(1-7)/Mas axis.

## 3. Discussion

In the present study, we explored cardiorenal protection by monotherapy with SGLT2i (empagliflozin), ERA (atrasentan), or RAS blockade (ramipril) and by dual or triple therapy adding SGLT2i and/or ERA to RAS blockade in type 2 diabetic mice. In line with findings from clinical trials, SGLT2i improved glycemic control, ramipril decreased systolic BP, all three agents displayed evidence of kidney protection, and SGLT2i and RAS blockade were more effective than ERA in heart protection, supporting the potential clinical relevance of the findings. The main findings refer to the impact of triple therapy, which has not been formally studied in the clinic. Triple therapy did not add further impact on glycemia, BP, eGFR, albuminuria, and heart function over monotherapy or dual therapy. However, it preserved the effects of the individual components and further added to organ protection by decreasing cardiomyocyte hypertrophy, preserving podocyte number, and activating the intrarenal ACE2/Ang(1-7)/Mas protective RAS pathway to levels not achieved by monotherapy or dual therapy. Overall, these data support that triple therapy preserves the benefits provided by the individual components or dual combinations on cardiorenal risk factors, as well as on kidney and heart injury. In addition, triple therapy overcomes the low heart protection already described for treatment with ERA alone [19] and provides added benefit over monotherapy or dual therapy for key pathways of tissue damage such as the intrarenal ACE2/Ang(1-7)/Mas axis.

In line with previous studies [20,21], empagliflozin was the only treatment that reduced blood glucose alone or combined with ramipril or atrasentan. The current finding confirms that other studied drugs did not interfere with the glucose-lowering effect of SGLT2i. Surprisingly, empagliflozin-treated db/db mice showed increased total body fat and the same trend in body weight. Although SGLT2i produce body weight loss in patients [22,23], previous studies that tested SGLT2i in mice showed no change [20,24] or even an increase in body weight [21]. This paradoxical outcome is likely a result of an increased insulin secretion and lower insulin resistance in response to treatment [21,25,26]. Considering that db/db mice do not present satiety, the higher insulin secretion linked to a constant food intake improves glucose handling and may increase body weight and fat [21,26].

Although the db/db did not show differences in BP before initiating the treatments, ramipril was added to evaluate the interaction of atrasentan and empagliflozin with an ACEi, as both drug classes have been used in RCT on top of RAS blockade [3,11,27]. The combination of ramipril with empagliflozin or atrasentan also showed a synergic effect on BP reduction. The greater BP reduction achieved by combination therapy may contribute to the observed heart and kidney beneficial effects. Ramipril had the most potent BP-lowering effect, and empagliflozin or atrasentan alone had no effect on BP, in line with previous reports, where treatment with empagliflozin [25,26] or atrasentan did not lower BP [28].

In terms of renal protection, only dual therapies with empagliflozin/ramipril or triple therapy reduced diabetic glomerular hyperfiltration. Although empagliflozin monotherapy prevented glomerular hyperfiltration in type 1 diabetic mice [24,29], this was not observed in our type 2 diabetic mouse [30]. It is worth noticing that a larger decrease in GFR was observed in the treatment groups that achieved lower BP, supporting the beneficial effect of BP control on glomerular hyperfiltration and showing an add-on effect of empagliflozin on top of the RAS blockade in preventing hyperfiltration by the glomerular hemodynamic effects of the combination. The histological evaluation of the kidney also revealed protective effects of all the experimental drugs in preventing fibrosis. In previous studies in diabetic animal models, both empagliflozin and atrasentan reduced inflammatory and profibrotic pathways in the kidney [21,24,31]. In addition, kidney protection by both SGLT2i and ERAs involves reducing TGF-β1, collagen, NF-κB, or oxidative stress [32,33]. Further, all active treatments prevented podocyte loss to some extent except for atrasentan in monotherapy, which is consistent with previous reports [34,35]. As expected, triple-therapy-treated mice showed the most important podocyte recovery supporting a synergy of the drug combination. Currently, an ongoing randomized clinical trial is assessing the combination of SGLT2i plus ERA in diabetic kidney disease: Zibotentan and Dapagliflozin for the Treatment of CKD (ZENITH-CKD Trial), which will provide insights into the impact of the combination on major renal outcomes.

Regarding heart injury, echocardiography revealed incipient diastolic dysfunction in db/db mice. Empagliflozin alone or within dual or triple therapy improved protection against diastolic dysfunction over ramipril alone by improving both IVRT and LA diameter. Furthermore, triple therapy with empagliflozin, atrasentan, and ramipril was the only combination that reduced cardiomyocyte hypertrophy and also achieved a larger increase of the α-MHC/β-MHC ratio than ramipril alone. Thus, triple therapy prevents the myosin isoform shift that occurs in diabetes, improving myocardium kinetics [36,37]. These findings are aligned with previous studies demonstrating heart protection by the study drugs that decrease inflammatory, profibrotic, and oxidative stress pathways [38,39,40,41]. Although ERA monotherapy with atrasentan was previously reported to improve systolic function and protect from ischemia-reperfusion-induced dysfunction in rats with type 1 diabetes [39], these results are not aligned with results from clinical trials [10]. In our experimental setting, atrasentan displayed mild heart protective effects, which is closer to clinical experience, but did not detract from heart protection in combination therapeutic regimes.

Intrarenal RAS plays a key role in diabetic nephropathy [42]. In concordance with previous studies, kidney AGT expression was increased in diabetic rodents [18,43], indicating intrarenal RAS activation in a diabetic milieu. In addition, in observational studies, urine AGT elevation is an early marker of diabetic kidney injury and antedates chronic kidney disease [44]. Although in some studies SGLT2i reduced urinary AGT [43], we did not observe an important impact of active interventions on renal AGT expression or urine AGT levels. Renin gene and protein expression were also slightly increased in db/db mice, which has already been described in the diabetic setting [45]. Additionally, renin was further increased in the groups treated with ramipril in monotherapy or in combination with empagliflozin and/or atrasentan. This well-known effect [46,47] may be in part ascribed to two main mechanisms: (1) ACE inhibition resulting in the interruption of the negative feedback mechanisms and (2) lower BP causing reduced renal blood flow and stimulation of glomerular afferent arteriole’s baroreceptors [46]. Moreover, SGLT2i was previously reported to increase both intrarenal and plasma renin activity [21,48,49]. The latter effect is probably mediated by renal baroreceptor activation despite an increased sodium chloride flow to the macula densa [50]. In the current study, dual empagliflozin/ramipril or triple therapy (empagliflozin, atrasentan, ramipril) clearly increased intrarenal renin expression compared to ramipril alone. These findings suggest that combining SGLT2i and ACEi with or without ERA further increased the first RAS limiting step in the kidney and, likely, in plasma.

While activation of the classical RAS (yielding Ang II) promotes inflammation and fibrosis, the non-classical pathway is characterized by ACE2-mediated production of Ang(1-7) that activates the Mas receptor and protects from diabetic nephropathy [51,52,53]. An increased kidney ACE2/ACE ratio in db/db [54] is thought to represent an early compensatory effect that opposes the diabetes-mediated increase in tubular Ang II. Combined therapies with ramipril resulted in the highest kidney ACE2/ACE activity ratio. This implies that the increased kidney RAS activation in diabetic mice noted by increased AGT, and fueled by ramipril-induced renin production, was redirected towards the non-classical RAS pathway (ACE2/Ang(1-7)/Mas) by ramipril, but, above all, by triple therapy. The combined effect of triple therapy, further activating the intrarenal non-classical RAS above the impact of ramipril, is a key observation regarding mechanisms of add-on tissue protection by the combined therapy (Figure 7) that affects subsequent proinflammatory or profibrotic pathways such as TGF-β1, NF-κB, or oxidative stress involved in diabetic nephropathy progression.

As every study, this work is not exempt of limitations. First, for ethical reasons, the study was only powered to identify differences in GFR at the end of the treatment and not for all the secondary outcomes analyzed. In addition, the mice losses that occurred during the conduct of the study also reduced the power of these secondary outcomes. Although the analysis by factors (drugs) helped us overcome this problem, in some comparisons, the post hoc tests did not show a significant protective effect for the monotherapies that were already described in previous studies, such as for UACR. Second, as depicted in our study, the db/db is considered a model of type 2 DM with normal blood pressure. This fact has to be taken into account when translating our results to the DM2 patient population, which is mainly hypertensive. Third, although we monitored water intake during the conduct of the experiment, no specific procedure was employed for body water measurement to evaluate fluid retention.

In conclusion, in the present study we characterized for the first time the impact of combining empagliflozin and atrasentan on top of RAS blockade in a clinically relevant model of cardiorenal injury: the db/db mouse with type 2 diabetes. Triple therapy preserved the benefits of the individual components or dual combinations on cardiorenal risk factors and kidney and heart injury and provided added benefit by preserving podocyte density, reducing cardiomyocyte hypertrophy, and further activating the intrarenal ACE2/Ang(1-7)/Mas axis. The study supports the enhanced cardiorenal protection by triple combination therapy in DKD patients, highlights the possibilities of combining the three drug classes, and may guide the design and choice of outcomes in future clinical trials.

## 4. Materials and Methods

### 4.1. Animals and Experimental Design

Eight-week-old male leptin-receptor-deficient diabetic mice (db/db) and non-diabetic heterozygous littermates (db/+) were purchased from Charles River (BKS.Cg-Dock7^m^+/+Lepr^db^J mice, strain code: 607). Mice were housed in an environmentally controlled room (22 ± 2 °C in a 12:12 h light-dark cycle) and had free access to standard chow and tap water. Twelve-week-old db/db mice were randomly assigned to 7 treatment arms for eight weeks: (A) db/db mice treated with vehicle (*n* = 15); (B) db/db mice treated with empagliflozin (*n* = 14); (C) db/db mice treated with empagliflozin and ramipril (*n* = 15); (D) triple therapy, db/db mice treated with empagliflozin, ramipril, and atrasentan (*n* = 15); (E) db/db mice treated with atrasentan (*n* = 8); (F) db/db mice treated with atrasentan and ramipril (*n* = 13); (G) db/db mice treated with ramipril (*n* = 9). Non-diabetic mice (db/m) treated with vehicle were included as non-diabetic controls (*n* = 12). Empagliflozin diluted in 0.5% hydroxyethyl-cellulose was administered daily by oral gavage (10 mg/Kg/day). Atrasentan (7 mg/kg/day) and ramipril (8 mg/kg/day) were administered through drinking water. Vehicle-treated mice received clean tap water and 0.5% hydroxyethyl-cellulose was administered daily by oral gavage. Body weight was monitored weekly, blood glucose biweekly, and blood pressure was measured before and after the 8 weeks of treatment. Animals were euthanized under pentobarbital anesthesia. Serum was collected by cardiac puncture. Kidneys and hearts were removed for molecular analysis (snap frozen) and histology (fixed in 10% formalin). The experimental design is displayed in Appendix A, whereas a scheme illustrating the mice excluded or deceased during the experiment is shown in Appendix A. The experimental protocol was approved by the institution Animal Ethics Committee (Project number: 47/18) and followed the European Council Directives for care of animals used for research (2010/63/EU).

### 4.2. Sample Size Calculation for the Experimental Procedure

The G*Power Software (V3.1.9.6, Germany) was used for sample size calculation. Identifying differences in GFR after 8 weeks of treatment was established as the primary outcome of the present study. To calculate sample size, we considered obtaining differences using one-way ANOVA test. A mean SD of 200-250μL/100g/min was considered between groups and a mean SD of 350-400μL/100g/min within groups, attending to data from previous studies [1,2]. Assuming a large effect size with an *f* value of 0.4 and expecting a 20% mortality, 15 mice were included in each db/db group.

### 4.3. Measured Outcomes

As previously stated, our main outcome was to detect beneficial effects of the study drugs against diabetic hyperfiltration and to identify if the combinations had a synergistic effect in GFR values measured at the end of the treatment. Our secondary objectives were to evaluate the protective effects of the study drugs against kidney injury (mesangial matrix expansion and podocyte density) and heart injury (cardiomyocyte hypertrophy and collagen deposition) at tissue level, and to describe the changes exerted by these drugs in the intrarenal RAS by assessing renin, ACE and ACE2 levels and/or activity.

### 4.4. Exclusion Criteria and Control of Confounding Factors

Exclusion criteria of the mice during the experimental procedure were: (1) death of the mouse before experimental endpoint, (2) absence of hyperglycemia (blood glucose <250mg/dL) in the db/db mice before starting the treatments (12 weeks of age), (3) development of malocclusion, and (4) confirmed infection of organs or tissues by direct visualization after euthanasia.

Regarding the control of the confounding factors, all mice were housed under the same conditions and exposed to the same number of procedures, including oral gavage that was performed during 8 weeks to all the mice included. Treatments were randomly assigned to each mice following a simple randomization. Researchers in charge of the experiment were aware of the group allocation except for the image acquisition (echocardiography, computed tomography and imaging of tissue sections) and analysis of the data.

### 4.5. Weight, Blood Glucose, and Blood Pressure Monitoring

Body weight was measured weekly from 9 weeks of age until the end of the experiment. Blood glucose was measured after 4 h of fasting before initiating treatment and every 2 weeks after treatment initiation using a hand-held glucose meter (Accu-Chek Performa, Roche, Basel, Switzerland). Non-invasive blood pressure (BP) was measured before starting treatment and after 8 weeks of treatment using the tail-cuff method (LE 5001, Harvard Apparatus, Holliston, MA, USA).

### 4.6. Computed Tomography Studies

Total body volume and body fat volume were measured by micro computed tomography before and after treatment. Studies were acquired with an equipment specifically designed for small animals (Quantum FX imaging system, Perkin Elmer, USA) and under isoflurane anaesthesia. Data were analysed by the Preclinical Imaging Platform staff at Vall d’Hebron Research Institute. Voxel size in obtained images was 3.2418 µm.^3^ Whole body volume was calculated performing a segmentation of complete body voxels from air voxels (minimum threshold −400 Hounsfield Units (HU)). Fat tissue was measured with a second segmentation of fat voxels (range −200–100 HU) [3,4]. Every segmentation was checked for proper fitting with the fat tissue. Finally, a local segmentation between intraabdominal and subcutaneous fat tissues was obtained applying a “seeds and frontiers” method. 

### 4.7. Transcutaneous Glomerular Filtration Rate

Glomerular filtration rate (GFR) was measured before and after treatment by transcutaneous measurement of fluorescein isothiocyanate (FITC)-sinistrin. Briefly, under isoflurane anesthesia, a side of the mouse’s back was shaved. In that portion of the skin, a fluorescent signal measuring device (Transdermal GFR monitor, MediBeacon, Mannheim, Germany) was attached with an adhesive patch and silk tape. After recording for 5 min the background signal of the skin, an intravenous bolus of a fluorescent agent (Fluorescein isothiocyanate (FITC) bounded sinistrin) was administered at a dose of 15 mg/100 g body weight. Then isoflurane anesthesia was stopped and the signal decay of FITC-sinistrin was recorded in conscious mice during one hour before removing the device. Data was analyzed with specific software (MPD Studio Version RC15, MediBeacon, Mannheim, Germany) using a 3-compartment model to obtain FITC-sinistrin half-life, which was converted to µL/100 g/min using a previously validated formula [55]. The difference in glomerular filtration rate (ΔGFR) was obtained subtracting the initial GFR to the final GFR in each mouse.

### 4.8. Albuminuria Measurement

First morning spot urine was obtained at the end of the experiment using an in-house urine multicollector for rodents (Patent number U202131356 (ZBM reference U691ES00), Spanish Patents and Trades Office). Urine albumin and creatinine levels were measured using commercial assays (Albuwell M, Ethos Biosciences, Logan Township, NJ, USA, and The Creatinine Companion, Ethos Biosciences, Logan Township, NJ, USA) to calculate the albumin-to-creatinine ratio.

### 4.9. Urine Glucose Measurement

Urine glucose was measured in spot urine samples obtained at the end of the experiment using the Mouse Glucose Assay kit (81692, Crystal Chem, Elk Grove Village, IL, USA) following manufacturer’s instructions. db/m urine samples were tested undiluted, whereas db/db urine samples were previously diluted 1:100 in double-distilled water.

### 4.10. Assessment of Heart Function

Echocardiography (Vivid IQ and L8-18i-D Linear Array 5-15MHz, General Electric Healthcare, Horten, Norway) was performed before and after the experiment. Following previous protocols [5], studies were carried out by a single operator who was blinded to the animal groups. Animals were anesthetized with 2% isoflurane and then placed in supine position on a heating pad to maintain body temperature. Ultrasound gel was applied on the left hemi-thorax and hearts were imaged in parasternal long- and short-axis projections. The following echocardiographic dimensions were measured: left atrium (LA) diameter, left ventricular (LV) end-diastolic diameter and LV end-systolic diameter. Systolic function was assessed by LV fractional shortening (FS, calculated by the formula (LVDd–LVDs)/LVDd x 100%), and ejection fraction (EF, calculated by the formula packed in the GE Healthcare Ultrasound Vivid 7 system). Diastolic function was assessed by the isovolumic relaxation time (IVRT). For all measurements, values were taken from the average of 3 consecutive cardiac cycles.

### 4.11. Glomerular Mesangial Matrix Expansion Measurement

Mesangial matrix expansion was evaluated in 10 representative cortical glomeruli in Periodic Acid Schiff (PAS) stained kidney sections. Paraffin blocks were cut at 4 µm, deparaffinized in xylene, and rehydrated through graded alcohols. Kidney sections were stained following PAS staining protocol and using Schiff’s reagent (3952016, MilliporeSigma, Darmstadt, Germany). Counterstain was performed with Haematoxylin Gill Nº3 solution (GHS332, MilliporeSigma, Darmstadt, Germany). Mesangial matrix area was adjusted in each 400× magnification image to tuft area. All histology images in the study were analyzed by two independent blinded observers using ImageJ (v1.53a, National Institutes of Health, Bethesda, MD, USA).

### 4.12. Podocyte Density Assessment

Wilms Tumor 1 (WT1) was detected by immunohistochemistry to count the number of podocytes per glomerular area. Paraffin blocks were cut at 4 µm, deparaffinized in xylene and rehydrated through graded alcohols. After deparaffinization, endogenous peroxidases were blocked in an aqueous solution containing 3% H_2_O_2_ and 10% methanol. Antigen retrieval was performed boiling the samples in citrate buffer (10 mM citric acid pH 6.0). The sections were then blocked in bovine serum albumin (5%), and incubated during 16 h at 4 °C with anti-WT1 primary antibodies (Dilution 1:100, ab89901, Abcam, Cambridge, UK). Biotinylated antibodies against rabbit IgGs (Dilution 1:250, BA-9500, Vector Laboratories, Burlingame, CA, USA) were employed as secondary antibodies. Proteins were visualized using the Avidin-Biotin Complex (ABC) Peroxidase Standard Staining Kit (32020, ThermoFisher Scientific, Waltham, MA, USA) followed by 3,3′-Diaminobenzidine (DAB) Enhanced Liquid Substrate System (D3939, MilliporeSigma, Darmstadt, Germany). Counterstaining was done with Haematoxylin Gill Nº3 solution. Podocyte density was evaluated in 20 representative cortical glomeruli and expressed as number of podocytes per mm^2^ of glomerular area.

### 4.13. Renin Detection by Immunohistochemistry in Kidney

Renin protein expression was assessed by immunohistochemistry. Paraffin blocks were cut at 4 µm, deparaffinized in xylene, and rehydrated through graded alcohols. After deparaffinization, endogenous peroxidases were blocked in an aqueous solution containing 3% H_2_O_2_ and 10% methanol. Antigen retrieval was performed boiling the samples in citrate buffer (10 mM citric acid pH 6.0). The sections were then blocked in bovine serum albumin (5%) and incubated during 16 h at 4 °C with anti-renin primary antibodies (Dilution 1:2000, ab212197, Abcam, Cambridge, UK). Biotinylated antibodies against rabbit IgGs (Dilution 1:250, BA-9500, Vector Laboratories, Burlingame, CA, USA) were employed as secondary antibodies. Proteins were visualized using the Avidin-Biotin Complex (ABC) Peroxidase Standard Staining Kit (32020, ThermoFisher Scientific, Waltham, MA, USA) followed by 3,3′-Diaminobenzidine (DAB) Enhanced Liquid Substrate System (D3939, MilliporeSigma, Darmstadt, Germany). Counterstaining was done with Haematoxylin Gill Nº3 solution. Once stained, 10 representative images of the juxtameglomerular apparatus per section were taken at 400× magnification to measure renin stained area.

### 4.14. Collagen Deposition Assessment in the Kidney and Heart by Picrosirius Red

Tubulointerstitial and myocardial collagen deposition was evaluated in Picrosirius-red (PSR)-stained sections. After deparaffinization and rehydration, sections were first counterstained during 15′ with 2.5 mg/mL Fast Green (11443054, Fisher Bioreagents, Altrincham, UK) and diluted in 1% acetic acid. Afterwards, they were washed and stained during 1 h with 0.1% Sirius Red (F3B) C.I.35782 (365548, MilliporeSigma, Darmstadt, Germany) and diluted in saturated picric acid solution (P6744, MilliporeSigma, Darmstadt, Germany). Ten different representative LV microphotographs were obtained at 400× magnification with an Olympus BX61 microscope. Fibrosis was quantified by two independent blinded observers as the percentage of collagen deposition with an automated color recognition processing plugin from the ImageJ analysis software (v1.53a, National Institutes of Health, Bethesda, MD, USA). Perivascular and endocardial collagen was excluded from heart measurements.

### 4.15. Cardiomyocyte Area Measurement in Hematoxylin–Eosin (H-E)-Stained Sections

Left ventricle (LV) cardiomyocyte hypertrophy was assessed in H-E-stained sections by measuring the area of 30 transversally sectioned cardiomyocytes. Paraffin blocks were cut at 4 µm, deparaffinized in xylene and rehydrated through graded alcohols. H-E staining in heart was performed following a modified procedure of the Diabetic Complications Consortium [56]. Eosin Y-solution 0.5% (109844, MilliporeSigma, Darmstadt, Germany) and Haematoxylin Gill Nº3 solution (GHS332, MilliporeSigma, Darmstadt, Germany) were employed for the procedure.

### 4.16. Gene Expression

Gene expression was analyzed by real-time qPCR using specific primers. Total RNA was isolated from kidney cortex or left ventricle by the phenol-guanidine thiocyanate-chloroform based method following the manufacturer instructions (TRItidy G, A4051, AppliChem GmbH, Dramstadt, Germany). RNA was retrotranscribed to first-strand cDNA using the High-Capacity RNA-to-cDNA Kit (4387406, ThermoFisher Scientific, Waltham, MA, USA). The amount of RNA used for cDNA synthesis was 1 µg. Each gene mRNA expression was analyzed by real-time PCR using PowerUp SYBR Green Master Mix (A25743, ThermoFisher Scientific, Waltham, MA, USA) and performed in the 7900HT Fast Real-Time PCR System (ThermoFisher Scientific, Waltham, MA, USA). For relative quantification, the ΔΔCT method was applied using the hypoxanthine-guanine phosphoribosyltransferase 1(Hprt1) as housekeeping gene. Vehicle-treated non-diabetic db/m were used as reference group. The primer sequences and the dilutions of 1 µg of cDNA are shown in Appendix A.

### 4.17. Angiotensinogen Measurement in Urine and Plasma

Angiotensinogen levels were assessed in urine and serum after treatments using a commercial ELISA kit (Mouse angiotensinogen (aGT) ELISA Kit, CSB-E08566m, Cusabio, Houston, TX, USA) following manufacturer instructions. A previous dilution of urine (1:4) and plasma (1:500) samples in a specific diluent provided by the kit was performed before testing.

### 4.18. Kidney Protein Extraction

Total kidney protein was extracted from kidney cortex and was homogenized in lysis buffer (50 mM HEPES pH 7.4, 150 mM NaCl, 0.5% Triton X-100, 0.025 mM ZnCl_2_, 0.1 mM Pefabloc SC Plus (11873601001, Roche, Ludwigsburg, Germany) and 1:7 dilution of EDTA-free protease inhibitor cocktail (11836170001, Roche, Ludwigsburg, Germany). Afterwards, the samples were centrifuged (16000RCF, 60 min, 4 °C) and the supernatant was recovered. Protein concentration was measured using the BCA protein assay kit (23225, ThermoFisher Scientific, Waltham, MA, USA) following the manufacturer’s protocol. These protein extracts were employed for ACE and ACE2 activity measurement.

### 4.19. ACE Activity

The determination of ACE activity was performed using a hippuric acid substrate bound to a histidine residue and a leucine residue (N-Hippuryl-L-Histidyl-L-Leucine) as previously described [57]. The renal cortical extracts, previously quantified, were adjusted to a concentration of 0.5 μg/μL with the same lysis buffer. A total of 2 µL of each of the samples were incubated for 25 min and at 37 °C with 73 µL of the test solution (Sodium borate buffer pH 8.3 (0.4 M Boric Acid, 0.9 M NaCl) with mM N-Hippuryl-L-Histidyl-L-Leucine (H1635, MilliporeSigma, Darmstadt, Germany). Then, the reaction was stopped by adding 180 μL of 0.28 M Sodium Hydroxide. After stopping the reaction, 15 μL of o-Phthalaldehyde (20 mg/mL in Methanol) were added to each sample and samples were incubated for 10 min at room temperature and protected from light. This second reaction was stopped by adding 30 µL of 3 N Hydrochloric Acid to the samples. Samples were then centrifuged for 5 min at 800 RCF. After that 200 µL of the supernatants from each sample were transferred to 96-well black plates and fluorescence (Excitation 360 nm and Emission 485 nm) was measured using the Varioskan LUX Multimode Microplate Reader (ThermoFisher Scientific, Waltham, MA, USA). Samples were carried out in duplicate. In addition, two wells with 75 ng/μL of recombinant human recombinant ACE (SAE0075, MilliporeSigma, Darmstadt, Germany) and two wells with lysis buffer alone were included as positive and negative control, respectively. Raw data was measured as relative fluorescence units (RFU) per mg and per h (RFU/mg/h).

### 4.20. ACE2 Activity

ACE2 activity was determined via a fluorescent enzymatic assay using an ACE2-quenched fluorogenic substrate (Mca-Ala-Pro-Lys(Dnp)-OH, BML-P163-0001, Enzo Life Sciences, Farmingdale, NY, USA) as previously described [57]. The assays were performed in black 96-well plates. In each well, 0.25 μg of kidney protein was added. Samples were carried out in duplicate. In addition, two wells with 0.008 ng/μL of recombinant mouse ACE2 (3437-ZN-010, R&D Systems, Minneapolis, MN, USA) and two wells with lysis buffer alone were included as positive and negative control, respectively. The reaction was initiated by addition of 50 μL of the substrate. The plates were incubated at 37 °C for 4 h and fluorescence (Excitation 320 nm and Emission 400 nm) was detected using the Varioskan LUX Multimode Microplate Reader (ThermoFisher Scientific, Waltham, MA, USA). Raw data was measured as relative fluorescence units (RFU) per mg and per h (RFU/mg/h).

### 4.21. Statistical Analysis

Data analysis was performed with Stata (V15.1, StataCorp LLC, College Station, TX, USA). Data are shown in tables as median and interquartile range (IQR) or in box plots. Outliers according to Tukey’s criteria were excluded from the analysis, although they are displayed in the box plots. A factorial ANOVA including diabetes’, empagliflozin’s, ramipril’s, and atrasentan’s main effects plus empagliflozin:ramipril and atrasentan:ramipril interaction was performed in the first place. When residuals of the ANOVA model did not follow a normal distribution, a bootstrapping method (100 replications) was employed to increase the accuracy of the model. After ANOVA, post hoc multiple comparisons were performed using Student’s t-test when data followed a normal distribution, and Mann–Whitney test when data did not follow a normal distribution. The *p* values were corrected afterwards using Sidak’s method. Appendix A shows the number of mice included in each of the measurements. A *p* value ≤ 0.05 was considered significant.

## 5. Patents

During the conduct of the study, an in-house urine multicollector for rodents was developed (Patent number U202131356 (ZBM reference U691ES00), Spanish Patents and Trades Office).

## Figures and Tables

**Figure 1 ijms-23-12823-f001:**
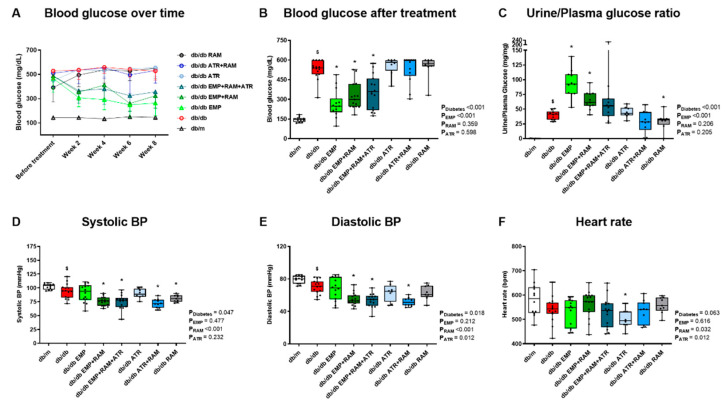
**Blood glucose, urine glucose, blood pressure (BP), and heart rate of vehicle-treated db/m mice, vehicle-treated db/db mice, and db/db mice treated with empagliflozin, atrasentan, ramipril, or their combinations**. (**A**) Mean blood glucose over time in mg/dL. Error bars display standard deviation (SD). (**B**) Blood glucose (mg/dL) after 8 weeks of treatment. (**C**) Urine glucose to plasma glucose ratio at the end of the treatment. (**D**) Systolic blood pressure in mmHg. (**E**) Diastolic blood pressure in mmHg. (**F**) Heart rate in bpm. **db/m**: non-diabetic mice treated with vehicle. **db/db**: diabetic mice treated with vehicle. **db/db EMP**: diabetic mice treated with empagliflozin. **db/db EMP + RAM**: diabetic mice treated with empagliflozin and ramipril. **db/db EMP + RAM + ATR**: diabetic mice treated with empagliflozin, ramipril, and atrasentan. **db/db ATR**: diabetic mice treated with atrasentan. **db/db ATR + RAM**: diabetic mice treated with atrasentan and ramipril. **db/db RAM**: diabetic mice treated with ramipril. Factorial ANOVA main effect results are displayed next to the graph. **P_Diabetes_**: diabetes’ main effect. **P_EMP_**: empagliflozin’s main effect. **P_RAM_**: ramipril’s main effect. **P_ATR_**: atrasentan’s main effect. **^$^** *p* < 0.05 vehicle-treated db/db mice vs. vehicle-treated db/m mice. *****
*p* < 0.05 any treated db/db mice compared with vehicle-treated db/db mice.

**Figure 2 ijms-23-12823-f002:**
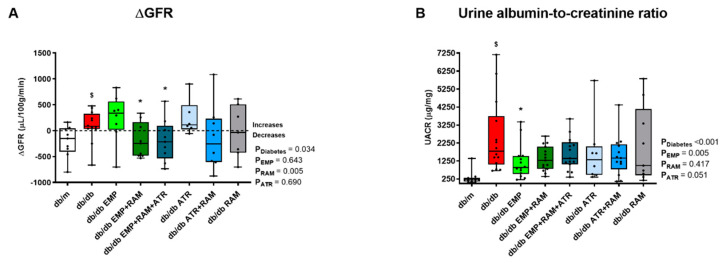
**Glomerular filtration rate difference (ΔGFR) from baseline and urinary albumin-to-creatinine ratio (UACR) at the end of the experiment in vehicle-treated db/m mice, vehicle-treated db/db mice, and db/db mice treated with empagliflozin, atrasentan, ramipril, or their combinations.** (**A**) ΔGFR measured in µL/100 g body weight/min. Difference was calculated subtracting pre-treatment GFR to post-treatment GFR in each mouse. (**B**) UACR at the end of the experiment. **db/m**: non-diabetic mice treated with vehicle. **db/db**: diabetic mice treated with vehicle. **db/db EMP**: diabetic mice treated with empagliflozin. **db/db EMP + RAM**: diabetic mice treated with empagliflozin and ramipril. **db/db EMP + RAM + ATR**: diabetic mice treated with empagliflozin, ramipril, and atrasentan. **db/db ATR**: diabetic mice treated with atrasentan. **db/db ATR + RAM**: diabetic mice treated with atrasentan and ramipril. **db/db RAM**: diabetic mice treated with ramipril. Factorial ANOVA main effect results are displayed next to the graph. **P_Diabetes_**: diabetes’ main effect. **P_EMP_**: empagliflozin’s main effect. **P_RAM_**: ramipril’s main effect. **P_ATR_**: atrasentan’s main effect. **^$^** *p* < 0.05 vehicle-treated db/db mice vs. vehicle-treated db/m mice. ***** *p* < 0.05 any treated db/db mice compared with vehicle-treated db/db mice.

**Figure 3 ijms-23-12823-f003:**
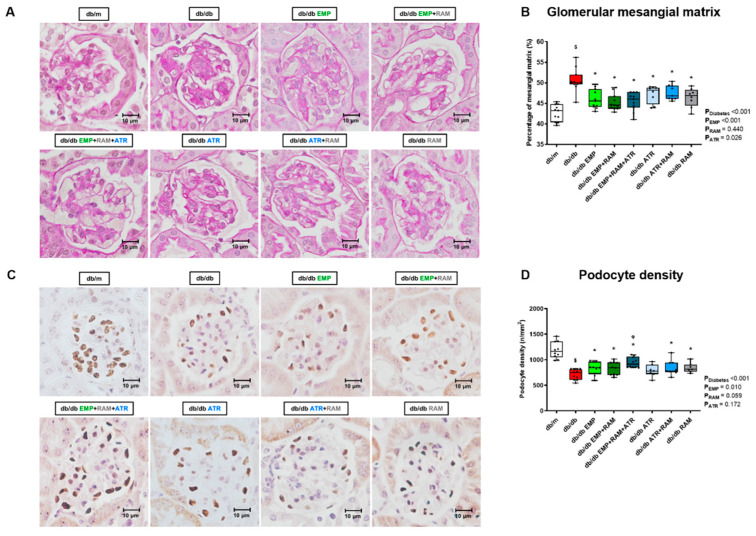
**Glomerular mesangial matrix area and podocyte density in vehicle-treated db/m mice, vehicle-treated db/db mice, and db/db mice treated with empagliflozin, atrasentan, ramipril, or their combinations**. (**A**) Representative PAS-stained glomerular microphotographs depicting glomerular mesangial matrix expansion (×400 magnification). (**B**) Percentage of glomerular mesangial matrix with respect to total glomerular tuft area. (**C**) Representative Wilms tumor 1 stained glomerular microphotographs showing podocyte density in each experimental group (×400 magnification). (**D**) Podocyte number per mm^2^ of glomerular area. **db/m**: non-diabetic mice treated with vehicle. **db/db**: diabetic mice treated with vehicle. **db/db EMP**: diabetic mice treated with empagliflozin. **db/db EMP + RAM**: diabetic mice treated with empagliflozin and ramipril. **db/db EMP + RAM + ATR**: diabetic mice treated with empagliflozin, ramipril, and atrasentan. **db/db ATR**: diabetic mice treated with atrasentan. **db/db ATR + RAM**: diabetic mice treated with atrasentan and ramipril. **db/db RAM**: diabetic mice treated with ramipril. Factorial ANOVA main effect results are displayed next to the graph. **P_Diabetes_**: diabetes’ main effect. **P_EMP_**: empagliflozin’s main effect. **P_RAM_**: ramipril’s main effect. **P_ATR_**: atrasentan’s main effect. **^$^** *p* < 0.05 vehicle-treated db/db mice vs. vehicle-treated db/m mice. ***** *p* < 0.05 any treated db/db mice compared with vehicle-treated db/db mice. **^φ^** *p* < 0.05 combinations of empagliflozin or atrasentan with ramipril db/db vs. ramipril db/db.

**Figure 4 ijms-23-12823-f004:**
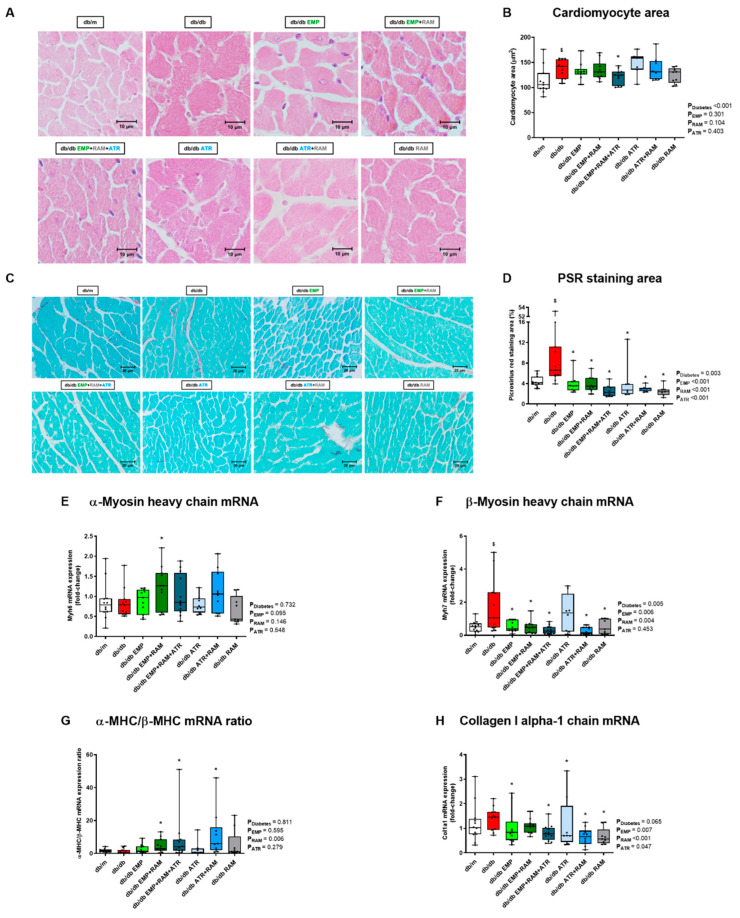
**Left ventricle cardiomyocyte hypertrophy, fibrosis, and gene expression of α-MHC, β-MHC and collagen I alpha-1 chain in vehicle-treated db/m mice, vehicle-treated db/db mice, and db/db mice treated with empagliflozin, atrasentan, ramipril, or their combinations**. (**A**) Representative hematoxylin-eosin microphotographs of transversally sectioned cardiomyocytes (×400 magnification). (**B**) Cardiomyocyte transversal area in µm^2^ measured in H&E-stained sections. (**C**) Representative Picrosirius-red-stained left ventricle sections (×400 magnification). (**D**) Left ventricle Picrosirius-red-stained area in µm^2^. (**E**) mRNA expression of α-myosin heavy-chain (α-MHC). (**F**) mRNA expression of β-myosin heavy-chain (β-MHC). (**G**) α-MHC/β-MHC ratio. (**H**) mRNA expression of collagen I alpha-1 chain. An almost significant increase in collagen I expression was observed in the vehicle-treated db/db group vs. the **db/m**: non-diabetic mice treated with vehicle. **db/db**: diabetic mice treated with vehicle. **db/db EMP**: diabetic mice treated with empagliflozin. **db/db EMP + RAM**: diabetic mice treated with empagliflozin and ramipril. **db/db EMP + RAM + ATR**: diabetic mice treated with empagliflozin, ramipril, and atrasentan. **db/db ATR**: diabetic mice treated with atrasentan. **db/db ATR + RAM**: diabetic mice treated with atrasentan and ramipril. **db/db RAM**: diabetic mice treated with ramipril. Factorial ANOVA main effect results are displayed next to the graph. **P_Diabetes_**: diabetes’ main effect. **P_EMP_**: empagliflozin’s main effect. **P_RAM_**: ramipril’s main effect. **P_ATR_**: atrasentan’s main effect. **^$^** *p* < 0.05 vehicle-treated db/db mice vs. vehicle-treated db/m mice. ***** *p* < 0.05 any treated db/db mice compared with vehicle-treated db/db mice.

**Figure 5 ijms-23-12823-f005:**
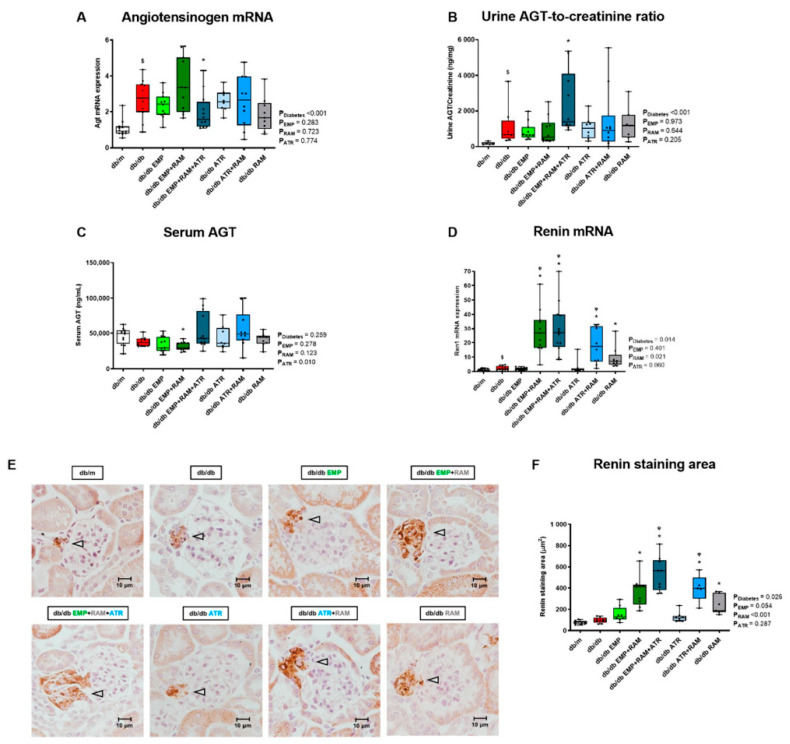
**Kidney, urine, and serum angiotensinogen and renal renin levels in vehicle-treated db/m mice, vehicle-treated db/db mice, and db/db mice treated with empagliflozin, atrasentan, ramipril, or their combinations.** (**A**) Angiotensinogen (AGT) mRNA expression. (**B**) Urine AGT-to-creatinine ratio. (**C**) Serum AGT values. (**D**) Renin mRNA expression. (**E**) Representative kidney sections showing specific renin staining (×400 magnification). Increased juxtaglomerular cell staining (grey arrows) is observed in treatments with ramipril and or ramipril combination with empagliflozin and/or atrasentan. (**F**) Renin stained area in µm^2^. **db/m**: non-diabetic mice treated with vehicle. **db/db**: diabetic mice treated with vehicle. **db/db EMP**: diabetic mice treated with empagliflozin. **db/db EMP + RAM**: diabetic mice treated with empagliflozin and ramipril. **db/db EMP + RAM + ATR**: diabetic mice treated with empagliflozin, ramipril, and atrasentan. **db/db ATR**: diabetic mice treated with atrasentan. **db/db ATR + RAM**: diabetic mice treated with atrasentan and ramipril. **db/db RAM**: diabetic mice treated with ramipril. Factorial ANOVA main effect results are displayed next to the graph. **P_Diabetes_**: diabetes’ main effect. **P_EMP_**: empagliflozin’s main effect. **P_RAM_**: ramipril’s main effect. **P_ATR_**: atrasentan’s main effect. **^$^** *p* < 0.05 vehicle-treated db/db mice vs. vehicle-treated db/m mice. ***** *p* < 0.05 any treated db/db mice compared with vehicle-treated db/db mice. **^φ^** *p* < 0.05 combinations of empagliflozin or atrasentan with ramipril db/db vs. ramipril db/db.

**Figure 6 ijms-23-12823-f006:**
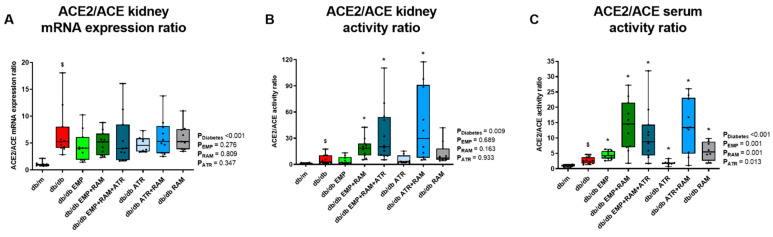
**Kidney ACE2/ACE mRNA expression and enzymatic activity ratio.** (**A**) ACE2/ACE mRNA expression ratio. (**B**) ACE2/ACE kidney activity ratio. (**C**) ACE2/ACE serum activity ratio. **db/m**: non-diabetic mice treated with vehicle. **db/db**: diabetic mice treated with vehicle. **db/db EMP**: diabetic mice treated with empagliflozin. **db/db EMP + RAM**: diabetic mice treated with empagliflozin and ramipril. **db/db EMP + RAM + ATR**: diabetic mice treated with empagliflozin, ramipril, and atrasentan. **db/db ATR**: diabetic mice treated with atrasentan. **db/db ATR + RAM**: diabetic mice treated with atrasentan and ramipril. **db/db RAM**: diabetic mice treated with ramipril. Factorial ANOVA main effect results are displayed next to the graph. **P_Diabetes_**: diabetes’ main effect. **P_EMP_**: empagliflozin’s main effect. **P_RAM_**: ramipril’s main effect. **P_ATR_**: atrasentan’s main effect. **^$^** *p* < 0.05 vehicle-treated db/db mice vs. vehicle-treated db/m mice. ***** *p* < 0.05 any treated db/db mice compared with vehicle-treated db/db mice.

**Figure 7 ijms-23-12823-f007:**
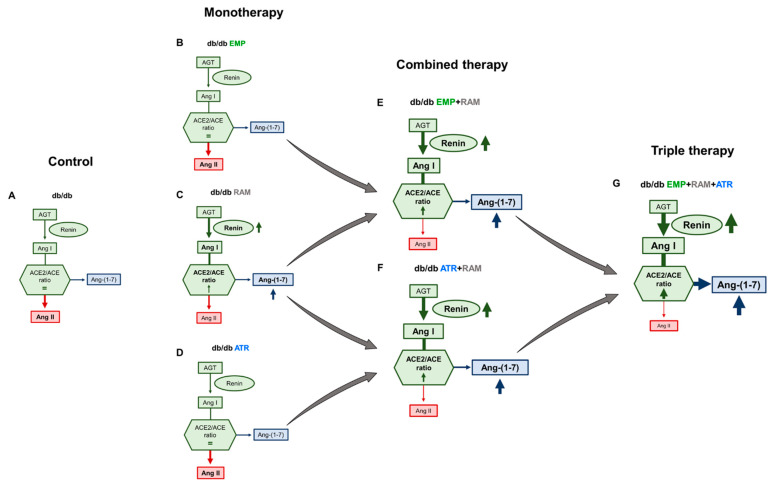
**Proposed modulation of intrarenal renin angiotensin system (RAS) in db/db mice treated with empagliflozin, atrasentan, ramipril, or their combination.** Treatment with empagliflozin (**B**) or atrasentan (**D**) alone did not significantly change intrarenal RAS but ramipril alone increases kidney renin levels (**C**) when compared to vehicle-treated db/db mice (**A**). However, either empagliflozin or atrasentan combined with ramipril ((**E**,**F**), respectively) increased renin activity and directed the intrarenal RAS towards the Ang (1–7) protective axis. These effects were highest in the combination therapy of empagliflozin, atrasentan and ramipril (**G**). **Ang I**: angiotensin I. **Ang II**: angiotensin II. **Ang (1-7)**: angiotensin (1-7). **ACE**: angiotensin converting enzyme. **ACE2**: angiotensin-converting enzyme 2. **db/db**: diabetic mice treated with vehicle. **db/db EMP**: diabetic mice treated with empagliflozin. **db/db EMP + RAM**: diabetic mice treated with empagliflozin and ramipril. **db/db EMP + RAM + ATR**: diabetic mice treated with empagliflozin, ramipril, and atrasentan. **db/db ATR**: diabetic mice treated with atrasentan. **db/db ATR + RAM**: diabetic mice treated with atrasentan and ramipril. **db/db RAM**: diabetic mice treated with ramipril.

**Table 1 ijms-23-12823-t001:** Body weight, body fat volume, and fat distribution at the end of the experiment of vehicle-treated db/m mice, vehicle-treated db/db mice, and db/db mice treated with empagliflozin, atrasentan, ramipril, or their combinations.

	db/m	db/db	db/db EMP	db/db EMP + RAM	db/db EMP + RAM + ATR	db/db ATR	db/db ATR + RAM	db/db RAM
Body weight (g)	28.6 (26.7–29.9)	41.7 ^$^ (36.3–46.6)	42.9 (36.8–46.0)	40.8 (36.0–46.7)	44.4 (39.8–49.4)	40.9 (36.9–45.0)	36.1 * (31.3–40.6)	40.9 (37.6–43.0)
Body fat volume (cm^3^)	3.3 (1.9–4.9)	18.4 ^$^ (16.1–23.5)	22.0 (16.5–30.3)	19.1 (16.5–25.4)	25.4 (19.8–30.0)	19.4 (15.8–25.7)	21.4 (14.9–22.9)	19.3 (16.4–25.7)
Subcutaneous fat volume (cm^3^)	2.0 (1.3–3.4)	12.4 ^$^ (10.6–14.4)	14.1 (11.3–18.9)	12.3 (10.5–14.8)	15.9 * (13.3–18.0)	12.7 (9.9–15.7)	12.8 (9.5–14.4)	12.6 (11.6–16.7)
Intra-abdominal fat volume (cm^3^)	1.4 (0.6–1.7)	6.8 ^$^ (5.4–9.0)	8.5 (5.3–10.7)	7.0 (5.9–9.6)	9.0 * (6.6–11.1)	6.8 (5.9–10.0)	7.8 (5.4–8.9)	6.7 (5.1–9.7)

**db/m**: non-diabetic mice treated with vehicle. **db/db**: diabetic mice treated with vehicle. **db/db EMP**: diabetic mice treated with empagliflozin. **db/db EMP + RAM**: diabetic mice treated with empagliflozin and ramipril. **db/db EMP + RAM + ATR**: diabetic mice treated with empagliflozin, ramipril, and atrasentan. **db/db ATR**: diabetic mice treated with atrasentan. **db/db ATR + RAM**: diabetic mice treated with atrasentan and ramipril. **db/db RAM**: diabetic mice treated with ramipril. Values are shown as median and interquartile range (Q1–Q3). **^$^** *p* < 0.05 vehicle-treated db/db mice vs. vehicle-treated db/m mice. ***** *p* < 0.05 any treated db/db mice compared with vehicle-treated db/db mice.

**Table 2 ijms-23-12823-t002:** Echocardiographic measurements after 8 weeks of treatment in vehicle-treated db/m mice, vehicle-treated db/db mice, and db/db mice treated with empagliflozin, atrasentan, ramipril, or their combinations.

	db/m	db/db	db/db EMP	db/db EMP+RAM	db/db EMP+RAM+ATR	db/db ATR	db/db ATR+RAM	db/db RAM
LA diameter (mm)	1.83 (1.75–2.06)	2.29 ^$^ (2.15–2.49)	2.07 * (2.00–2.15)	2.09 * (2.06–2.15)	2.11 * (1.99–2.28)	2.29 (2.08–2.54)	2.16 (1.85–2.29)	2.49 (1.92–2.50)
LV end-diastolic diameter (mm)	3.63 (3.50–3.77)	3.61 (3.35–3.74)	3.61 (3.48–3.87)	3.65 (3.40–3.76)	3.58 (3.38–3.68)	3.77 (3.54–3.86)	3.53 (3.40–3.79)	3.49 (3.35–3.63)
LV end-systolic diameter (mm)	2.28 (2.17–2.34)	2.18 (2.05–2.44)	2.31 (1.88–2.42)	2.14 (2.04–2.46)	2.19 (2.03–2.27)	2.40 * (2.31–2.52)	2.12 (1.93–2.47)	2.14 (1.91–2.29)
IVS thickness (mm)	0.89 (0.85–0.93)	0.90 (0.87–0.98)	0.85 (0.79–0.98)	0.89 (0.81–0.92)	0.89 (0.82–0.92)	0.90 (0.85–0.95)	0.87 (0.80–0.89)	0.91 (0.89–0.91)
Ejection fraction (%)	72.1 (70.4–75.6)	73.1 (70.7–74.2)	75.2 (69.9–78.3)	74.7 (70.9–80.0)	75.4 (70.0–79.6)	71.2 (69.7–73.0)	76.7 (71.9–78.1)	74.2 (72.7–76.2)
IVRT (ms)	14.3 (12.5–17.8)	19.9 ^$^ (17.5–21.4)	15.1 * (13.3–16.8)	14.5 * (12.0–15.6)	13.0 * (11.5–16.4)	17.5 (14.5–24.8)	14.7 * (10.8–18.9)	17.7 (10.8–19.9)

**LA**: left atrium. **LV**: left ventricle. **IVS**: interventricular septum. **IVRT**: isovolumetric relaxation time. **db/m**: non-diabetic mice treated with vehicle. **db/db**: diabetic mice treated with vehicle. **db/db EMP**: diabetic mice treated with empagliflozin. **db/db EMP + RAM**: diabetic mice treated with empagliflozin and ramipril. **db/db EMP + RAM + ATR**: diabetic mice treated with empagliflozin, ramipril, and atrasentan. **db/db ATR**: diabetic mice treated with atrasentan. **db/db ATR + RAM**: diabetic mice treated with atrasentan and ramipril. **db/db RAM**: diabetic mice treated with ramipril. Values are shown as median and interquartile range (Q1–Q3). **^$^** *p* < 0.05 vehicle-treated db/db mice vs. vehicle-treated db/m mice. ***** *p* < 0.05 any treated db/db mice compared with vehicle-treated db/db mice.

## Data Availability

The datasets generated and/or analyzed during the current study are available from the corresponding authors on reasonable request.

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
