# Peer review of "Enhanced Cardiorenal Protective Effects of Combining SGLT2 Inhibition, Endothelin Receptor Antagonism and RAS Blockade in Type 2 Diabetic Mice"

_ijms, 2022, doi:10.3390/ijms232112823_

Round 1
Reviewer 1 Report
I only have one question about the administration route of the mice treated with Atrasentan and ramipril because it was done in the drinking water and it is unclear how the dose received by each mouse is controlled, because being a diabetic model it presents polydipsia and polyphagia, which would cause the animal to suffer an overdose, which could alter the pharmacological effects. Furthermore, the vehicle employed (water, saline solution, or other) and the commercial presentation of each of the drugs utilized are not indicated.
Author Response
We would like to thank the reviewers for their comments and suggestions. We have revised the manuscript entitled "Enhanced cardiorenal protective effects of combining SGLT2 inhibition, endothelin receptor antagonism and RAS blockade in type 2 diabetic mice" and we believe that the revised version of the manuscript has improved substantially. Following there is a detailed description and answer to the comments and changes proposed by the reviewer:
Reviewer 1
Comment 1, I only have one question about the administration route of the mice treated with atrasentan and ramipril because it was done in the drinking water and it is unclear how the dose received by each mouse is controlled, because being a diabetic model it presents polydipsia and polyphagia, which would cause the animal to suffer an overdose, which could alter the pharmacological effects. Furthermore, the vehicle employed (water, saline solution, or other) and the commercial presentation of each of the drugs utilized are not indicated.
Response, We thank the reviewer for his/her comment. When designing the experiment we chose the concentrations of the drugs in the drinking water based on previous manuscripts that already describe that water intake in db/db mice is usually 18-120 mL/mice/day (Gallo LA et al. Sci Rep, 2016, PMID 27226136). Therefore, polydipsia was already considered when choosing the doses. Moreover, to confirm that water intake was similar in our experiment to that described on the literature we measured it before starting the treatments and every 2 weeks until the end of the treatment. Final water intake (20 weeks of age) is already shown in Figure S3.
Regarding the vehicle, we employed 0.5% hydroxyethyl-cellulose for oral gavage that was mixed with empagliflozin for those that received active treatment. We have added all this information to the Methods section of the new version of the manuscript (see page 14).
Reviewer 2 Report
This is an interesting article aimed to evaluate the cardiorenal beneficial effects of the combination of SGLT2i and ERA on top of renin-angiotensin system blockade.
The introduction sets the scene by describing role of SGLT2i and ERA treatment in chronic kidney disease. The methodology is clear and concise and provide necessary information to repeat the method as well as critical steps. The figures outline the key message and results are well presented. However, there are some points in Discussion part need to be addressed:
1. How can you explain that only dual therapies with empagliflozin/ramipril or triple therapy reduce diabetic glomerular hyperfiltration?
2. How do you explain that ERA monotherapy with atrasentan that was previously reported to improve systolic function and protect from ischemia-reperfusion-induced dysfunction in rats with type 1 diabetes is not aligned with results from clinical trials?
3. Conclusions need to be highlighted.
Author Response
We would like to thank the reviewers for their comments and suggestions. We have revised the manuscript entitled "Enhanced cardiorenal protective effects of combining SGLT2 inhibition, endothelin receptor antagonism and RAS blockade in type 2 diabetic mice" and we believe that the revised version of the manuscript has improved substantially. Following there is a detailed description and answer to the comments and changes proposed by the reviewer:
Reviewer 2
This is an interesting article aimed to evaluate the cardiorenal beneficial effects of the combination of SGLT2i and ERA on top of renin-angiotensin system blockade.
The introduction sets the scene by describing role of SGLT2i and ERA treatment in chronic kidney disease. The methodology is clear and concise and provide necessary information to repeat the method as well as critical steps. The figures outline the key message and results are well presented. However, there are some points in Discussion part need to be addressed:
Comment 1, How can you explain that only dual therapies with empagliflozin/ramipril or triple therapy reduce diabetic glomerular hyperfiltration?
Response, We thank the reviewer for his/her comments. Empagliflozin on top of RAS blockade has shown to reduce hyperfiltration in randomized clinical trials. For that reason, we think that this drug in combination with ramipril prevents hyperfiltration by their dual hemodynamic mechanistic effect (vasoconstriction of the afferent glomerular arteriole and vasodilation of the efferent glomerular arteriole).
Comment 2, How do you explain that ERA monotherapy with atrasentan that was previously reported to improve systolic function and protect from ischemia-reperfusion-induced dysfunction in rats with type 1 diabetes is not aligned with results from clinical trials?
Response, We appreciate the reviewer's comment. Although atrasentan has shown heart protective effects in animal studies (Samad MA et al. PLoS One, 2015, PMID 25775254, and Kala P et al. J Hypertens, 2022, PMID 36204993), the results obtained in randomized controlled trials show the opposite, with fluid overload being a common adverse event (Mann JFE et al. J Am Soc Nephrol, 2010, PMID 20167702, and Heerspink HJL et al. Lancet, 2019, PMID 30995972). Moreover, in RCTs atrasentan was tested on top of RAS blockade. In our study, when combined with ramipril and empagliflozin, atrasentan shows greater cardiorenal protective effects with an increase in podocyte density, a reduction of cardiomyocyte hypertrophy or an improvement of the α-MHC/β-MHC ratio. However, atrasentan alone shows no protective effect, but it does not worsen renal, or heart injury as seen in RCTs. These results are in line with previous studies that showed no renal or heart effects for atrasentan in monotherapy (Ritter C et al. Kidney Blood Press Res, 2014, PMID 25300759). Following the reviewer's suggestion, we have added some of the references to the beginning of the Discussion section, where the limitations of atrasentan monotherapy were already mentioned (see page 11).
Comment 3, Conclusions need to be highlighted.
Response, We thank the reviewer for his/her kind comment. The enhanced cardiorenal protection of the triple therapy has been highlighted in the last paragraph of the Discussion section (see page 14).